# Characterizing collaborative transcription regulation with a graph-based deep learning approach

**Zhenhao Zhang**[1], **Fan Feng**[1], **Jie Liu**[1,2]*

**1** Department of Computational Medicine and Bioinformatics, University of Michigan, Ann Arbor, Michigan, United States of America, **2** Department of Computer Science and Engineering, University of Michigan, Ann Arbor, Michigan, United States of America

* drjieliu@umich.edu

## Abstract

Human epigenome and transcription activities have been characterized by a number of sequence-based deep learning approaches which only utilize the DNA sequences. However, transcription factors interact with each other, and their collaborative regulatory activities go beyond the linear DNA sequence. Therefore leveraging the informative 3D chromatin organization to investigate the collaborations among transcription factors is critical. We developed ECHO, a graph-based neural network, to predict chromatin features and characterize the collaboration among them by incorporating 3D chromatin organization from 200-bp high-resolution Micro-C contact maps. ECHO predicted 2,583 chromatin features with significantly higher average AUROC and AUPR than the best sequence-based model. We observed that chromatin contacts of different distances affected different types of chromatin features' prediction in diverse ways, suggesting complex and divergent collaborative regulatory mechanisms. Moreover, ECHO was interpretable via gradient-based attribution methods. The attributions on chromatin contacts identify important contacts relevant to chromatin features. The attributions on DNA sequences identify TF binding motifs and TF collaborative binding. Furthermore, combining the attributions on contacts and sequences reveals important sequence patterns in the neighborhood which are relevant to a target sequence's chromatin feature prediction.

## Author summary

Human transcription activities are regulated by chromatin features including transcription factor binding, histone modification, and DNase I hypersensitive site. Recently many computational models are proposed to predict chromatin features from DNA sequence. However, human genome has a complex and dynamic spatial organization, and chromatin loops form and bring regulatory elements that lie far apart on the genomic sequence into spatial proximity so that transcription factors which bind far apart may interact with each other. Therefore, to investigate the collaborations among chromatin features, utilizing 3D chromatin organization is critical. In this work, we propose a graph neural

**Data Availability Statement:** The chromatin contacts are obtained from 200-bp Micro-C contact maps of two cell lines, hESC and HFF. Our 2,583 chromatin feature profiles are collected from the Encyclopedia of DNA elements (ENCODE) and the

International Human Epigenome Consortium (IHEC). The TF binding profiles are all collected from ENCODE narrowPeak bed files. The download urls of the chromatin feature profiles and the source code of ECHO are available in the GitHub repository: https://github.com/liu-bioinfo-lab/echo.

**Funding:** ZZ, FF and JL were supported by National Human Genome Research Institute award R35HG011279. The funders had no role in study design, data collection and analysis, decision to publish, or preparation of the manuscript.

**Competing interests:** The authors have declared that no competing interests exist.

network model to predict chromatin features in the light of 200bp-resolution Micro-C contact maps which capture fine-scale chromatin contacts well. Furthermore, by interpreting the model, we identify important chromatin contacts which contribute to chromatin feature prediction, and characterize the collaborations among these chromatin features, which helps researchers understand transcription factor collaborative binding mechanisms.

## Introduction

The human body consists of hundreds of different cell types in spite of the identical genotype [1], and the precise gene expression, cell identities and plasticity are dictated by transcriptional regulatory mechanisms [2]. In this process, transcription factors (TFs) bind DNA regulatory sequences by recognizing their binding motifs and form a complex system that regulates chromatin and transcription [3]. Motivated by this, plenty of current deep learning based prediction models characterize TF binding and other chromatin features such as histone marks and DNase I hypersensitive sites,only from the DNA sequences. However, growing evidence indicates that transcription factors interact with each other [4, 5] and may form condensates in the 3D chromatin organization [6]. Therefore, it is critical for a computational model to properly characterize collaborative transcription regulation in the light of 3D chromatin organization.

Currently, multiple deep learning based models are proposed to predict chromatin features from DNA sequence, but most of these models predict chromatin features without utilizing the 3D chromatin organization. Here according to whether 3D chromatin organization is utilized, we categorize the deep learning based computational works for predicting chromatin features into *sequence-based* and *graph-based* models. Well-known sequence-based models, such as DeepSEA [7], DanQ [8], DeepBind [9], Basset [10], Basenji [11], and SATORI [12], predict chromatin features only from DNA sequences and ignore the informative chromatin structures. To the best of our knowledge, the only graph-based chromatin feature prediction model ChromeGCN [13] uses a gated graph convolution network to leverage the neighborhood information from 1kb resolution Hi-C contact maps which capture the spatial interactions between 1kb genomic regions, but it does not fully characterize cooperation among chromatin features.

In this paper, we proposed ECHO (Epigenomic feature analyzer with 3D CHromosome Organization), a graph-based neural network to predict the chromatin features and identify collaboration among them by including 3D chromatin organization. In ECHO, nucleosome-resolution Micro-C contact maps which capture higher resolution chromatin contacts than Hi-C, were represented as graphs, in which nodes were non-overlapping 200 base pair (bp) long genomic segments and weighted edges were chromatin contacts between these segments. Inspired by recent work [14, 15], we transformed the graph structure data to grid structure, which was then operated by 1D convolutions to leverage the neighborhood information. ECHO accurately predicted chromatin features including transcription factor binding, histone modifications, and DNase I hypersensitive sites, with an average AUROC 0.921 and an average AUPR 0.378 in the prediction of 2,583 chromatin features, significantly higher than the best *sequence-based* model with an average AUROC 0.885 and an average AUPR 0.318. In addition, by evaluating the model using contacts of different distances, we identified patterns about how chromatin contacts of different distances affected chromatin feature prediction.

Moreover, the contributions of Micro-C contacts and DNA sequences to the investigated chromatin features were characterized by applying an attribution method to ECHO. For TFs

with known motifs, the corresponding high attribution score regions on DNA sequences match their binding motifs. For TFs without known motifs, previous sequence-based methods may fail to extract prediction patterns from the binding DNA sequences. From the comparison of ECHO and sequence-based model's results, we found that these TFs' prediction accuracy improved more than that of TFs with known motifs. Therefore, ECHO leveraged chromatin structures and extracted information from the neighborhood to assist prediction. As we attributed the TFs to the neighborhood, the high attribution score regions in the neighbor sequences also matched other TFs' binding motifs, which indicated that ECHO recovered TF collaborative binding activities. Furthermore, important sequence patterns were revealed from high attribution score regions in the neighborhood, and some were consistent with existing biological knowledge.

## Results

### A graph neural network that leverages 3D chromatin organization and predicts chromatin features

We proposed ECHO, a graph neural network model to predict various chromatin features, including transcription factors (TFs) binding activities, histone modifications, and DNase I hypersensitive sites (DHSs), and characterize their collaboration in the light of high-resolution 3D chromatin organization. ECHO takes inputs including one-hot representations of 1000-bp DNA sequences from the reference genome and chromatin contacts from Micro-C contact maps, and outputs a vector of predicted chromatin features. The architecture of ECHO consists of sequence layers, graph layers, and one prediction layer (Fig 1a and S6 Fig). The sequence layers extract sequence features. The graph layers aggregate neighborhood information to extract features from the neighborhood. The prediction layer makes predictions from previous graph layers. Unlike previous sequence-based models such as DeepSEA [7] and DanQ [8], ECHO leveraged neighborhood information to assist the prediction of chromatin features. Unlike ChromeGCN [13] which used graph convolution networks (GCNs) and aggregated the neighborhood information based on the weighted adjacency matrix, we first transformed the graph structure of chromatin contact data to a grid structure [14, 15], and then we performed convolution to learn features from sequential and spatial chromatin structures. Additionally, ECHO utilized 200bp resolution Micro-C contact maps since many of the DHS peaks were narrower than 200bp (S5 Fig), and the widths of TF binding sites were typically much smaller than 200bp. To predict chromatin features at this high resolution, Micro-C which detected chromatin contacts between much shorter fragments than Hi-C and better illustrated fine-scale chromatin interactions, was more desirable. Moreover, different from ChromeGCN which required full batch training, ECHO used neighborhood sampling and mini-batch training which made it applicable to large graphs [16].

### ECHO predicted chromatin features more accurately than baseline methods

We first compared the prediction performance of ECHO and three sequence-based models, including DeepCNN (the deep neural network with six convolutional layers used in ExPecto framework [17]), DeepSEA [7], and DanQ [8]. AUROC and AUPR scores were calculated for individual chromatin features. Overall, ECHO predicted chromatin features accurately with an averaged AUROC of 0.921, which is significantly higher than DeepCNN (AUROC 0.885, *p*-value 6.73E−97), DeepSEA (AUROC 0.881, *p*-value 1.89E−124), and DanQ (AUROC 0.881, *p*-value 1.57E−122) (Fig 1b). Similarly, ECHO yielded an averaged AUPR of 0.378, which is

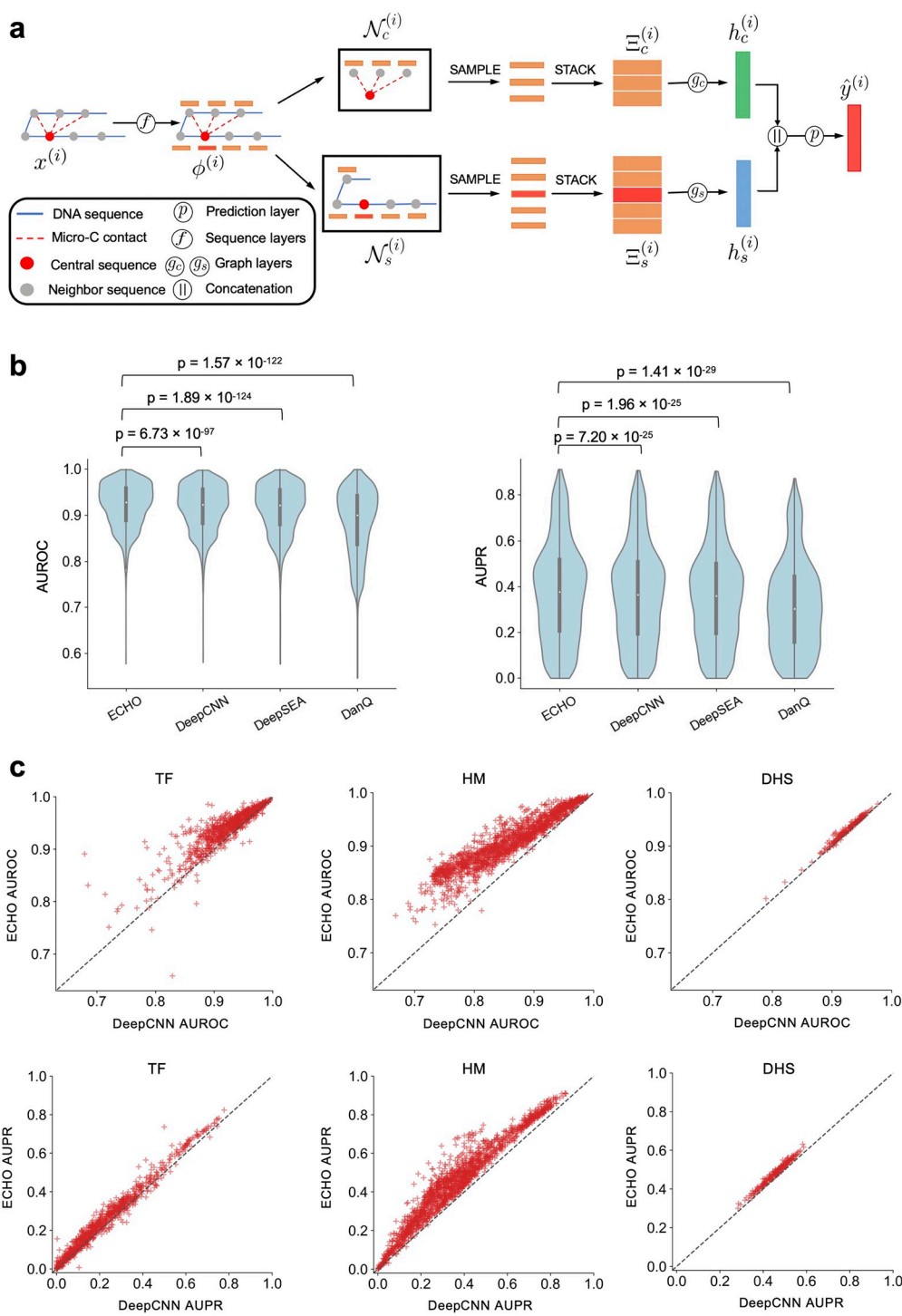

**Fig 1. Architectural details of ECHO and its empirical performance in chromatin feature prediction tasks.** (a) Model architecture. The model inputs are one-hot representations of DNA sequences. The inputs $x$ are first fed into sequence layers $f$ to extract hidden representations $\phi$. For central sequence $i$, we first sample a fixed number of sequential neighbor sequences from its sequential neighbor set $\mathcal{N}_c^{(i)}$ and a fixed number of spatial neighbor sequences from the spatial neighbor set $\mathcal{N}_s^{(i)}$. The hidden representations of sampled sequential neighbors are stacked into a feature matrix $\Xi_c^{(i)}$ which is input to graph layers $g_c$ to learn an updated hidden representation $h_c^{(i)}$. The feature matrix $\Xi_s^{(i)}$ stacked by spatial neighbors is fed into $g_s$ to learn an updated hidden representation $h_s^{(i)}$. A feature embedding concatenated by the two hidden representations will be fed into the final prediction layer $p$ to compute a chromatin

feature vector $y^{(i)}$. (b) The AUROC and AUPR scores for individual chromatin feature are provided according to ECHO and three baselines including DeepCNN, DeepSEA and DanQ. The given *p*-values from paired *t*-tests indicate that ECHO achieves a significant improvement. (c) Scatter plots of each chromatin feature's AUROC score (top panels) and AUPR scores (bottom panels) from ECHO and DeepCNN. ECHO outperforms DeepCNN on TF, histone mark and DHS profile prediction for both AUROC scores and AUPR scores.

significantly higher than DeepCNN (AUROC 0.318, *p*-value 7.20E−25), DeepSEA (AUROC 0.312, *p*-value 1.96E−25), and DanQ (AUROC 0.316, *p*-value 1.41E−29) (Fig 1b). When we separated model performance in terms of the three categories of chromatin features, namely TF, histone mark, and DHS, we observed that the ECHO's dramatically improved the prediction for histone mark, moderately improved the prediction for TF, and mildly improved the prediction for DHS (Fig 1c and S1 Fig).

ECHO used both spatial neighbors from the contact maps and sequential neighbors along the DNA sequence to predict chromatin features. To show the importance of spatial neighbors from contact maps, we built ECHO with only a large number of sequential neighbors along the DNA sequence but no spatial neighbors, and the model yielded an AUROC score 0.917 and an AUPR score 0.369 which were lower than those from original ECHO model (S2 Table). We also showed the importance of sequential neighbors along the DNA sequence by building ECHO with the spatial neighbors only, and the model achieved an AUROC of 0.918 and an AUPR of 0.372 which were sightly lower than those from original ECHO (S2 Table).

Since the collected chromatin features are from 402 cell lines, we further compared ECHO with DeepCNN in a cell-type specific chromatin feature prediction task. ECHO achieved higher mean AUC scores and mean AUPR scores for almost all the collected cell types (S2 and S3 Figs). Although the prediction performance improvement is not obvious for some under-represented cell types (i.e., these with only one collected chromatin feature, see S3 Fig), the improvement on other cell types is much more significant (S2 Fig).

The current results from ECHO used DeepCNN as the pre-train model to pre-train the sequence layers, and we were curious whether the improvement was consistent if we changed DeepCNN to DeepSEA and DanQ. Experiment results showed that when applying the same ECHO framework to different pre-train sequence-based models such DeepSEA and DanQ, the prediction was improved consistently, ECHO pre-trained by DeepSEA had an average AUROC of 0.918 and AUPR of 0.373, and ECHO pre-trained by DanQ yielded an average AUROC of 0.919 and AUPR of 0.386 (S2 Table).

In addition, we compared ECHO with a graph-based model ChromeGCN [13]. ChromeGCN's full batch training is computationally prohibitive on our evaluation datasets (over 2.9 million sequence segments and 2,583 chromatin features). Therefore, we used the small dataset processed by ChromeGCN to compare our ECHO and ChromeGCN. The small dataset was for GM12878 only, which included 103 chromatin features (90 TFs, 11 histone marks, and 2 DHSs) and a 1kb resolution Hi-C contact map. For both ECHO and ChromeGCN, each 2000-bp input DNA sequence and its reverse complement sequence were first embedded using one-hot encoding (use vectors of 0.25 for sequence gaps and unannotated regions). DeepCNN was selected as the pre-train model. In ECHO, 30 spatial and 10 sequential neighbors were sampled for each input sequence. In ChromeGCN, a combination of constant neighborhood and Hi-C contact maps was used. In both ChromeGCN and ECHO, hidden representations of each original sequence with its reverse complement were extracted when the pre-train model achieved a minimum loss on validation sets, which were then input to the graph layers. The outputs for the original sequence and its reverse complement were averaged in the calculation of the loss. The hyperparameters were all kept the same for ECHO and

ChromeGCN. ECHO predicted GM12878's chromatin features accurately with a mean AUROC of 0.924, a mean AUPR of 0.429 and a mean recall at 50% FDR of 0.399, which outperformed ChromeGCN (AUROC 0.916, AUPR 0.406 and recall at 50% FDR 0.372) (S3 Table and S4 Fig).

## The influences of Micro-C contact distances on chromatin feature prediction

We next investigate whether the improved chromatin feature prediction depends on the chromatin contact distances. To compare the influences of long-range and short-range Micro-C contacts on chromatin feature prediction, we separated the Micro-C contacts into four non-overlapping groups by contact distance within 0−1 kb, 1−5 kb, 5−20 kb, and over 20 kb. The four groups have 14, 30, 26, and 13 million chromatin contacts, respectively. For each group, we ran ECHO only using the specified range of Micro-C contacts without sampling sequential neighbors. The effects of each contact distance group were evaluated by the same model performance criteria, including AUROC and AUPR. By comparing the performance of each group with the baselines and original ECHO with all Micro-C contacts, we found both the long-range and short-range Micro-C contacts contributed to chromatin feature prediction. Overall, short-range Micro-C contacts produced a more significant improvement on model performance, which was indicated by the more significant *p*-values from the paired *t*-tests with baseline DeepCNN (*p*-values for AUROC: 0−1 kb (1.95E−29), 1−5 kb (1.91E−42), 5−20 kb (1.62E−13), 20− kb (3.25E−4); *p*-values for AUPR: 0−1 kb (1.03E−10), 1−5 kb (1.12E−10), 5−20 kb (3.81E−4), 20− kb (3.24E−2) Fig 2a).

We further investigated the influences of Micro-C contact distances on the prediction of TFs and histone marks separately. Surprisingly, three patterns were revealed (Fig 2b, 2c, 2d, and 2e). First, for predicting TFs related to chromatin structure maintenance such as CTCF, RAD21, and SMC3, ECHO using short-range Micro-C contacts with contact distances within 0−1 kb and ECHO using long-range Micro-C contacts with contact distances over 20 kb improved the model the most (Fig 2b). The high AUPR and AUROC scores for over 20 kb Micro-C contacts indicated that ECHO leveraged long-range interactions between these structural proteins. Second, for chromatin features mostly binding to promoters and enhancers and related to gene activation such as H3K4me3, H3K4me1, H3K27ac, H3K9ac, and POLR2A, ECHO achieved the highest AUROC and AUPR scores with 0−5 kb short-range Micro-C contacts, while the performance deteriorated as contact distance increased (Fig 2c and S7 Fig).

The last pattern was that the Micro-C contacts with contact distance within 1−20 kb (especially 1−5 kb) improved the prediction of gene elongation marks (including H3K36me3, H3K79me2, and H4K20me1, Fig 2d) and repressive marks (including H3K9me3 and H3K27me3, Fig 2e) the most. Gene elongation marks usually spanned longer than gene activation marks, resulting in correlation over longer distances. DNA methylation was found to be better correlated between distant loci in the B compartment where the repressive marks were enriched [18], and these correlation patterns were also revealed by ECHO. Overall, these patterns validated our assumption that Micro-C contacts of different contact distances could be related to chromatin features in different ways, and they indicated that ECHO identified these complex collaborative regulatory mechanisms, which might contribute to its great improvement on histone mark and TF prediction.

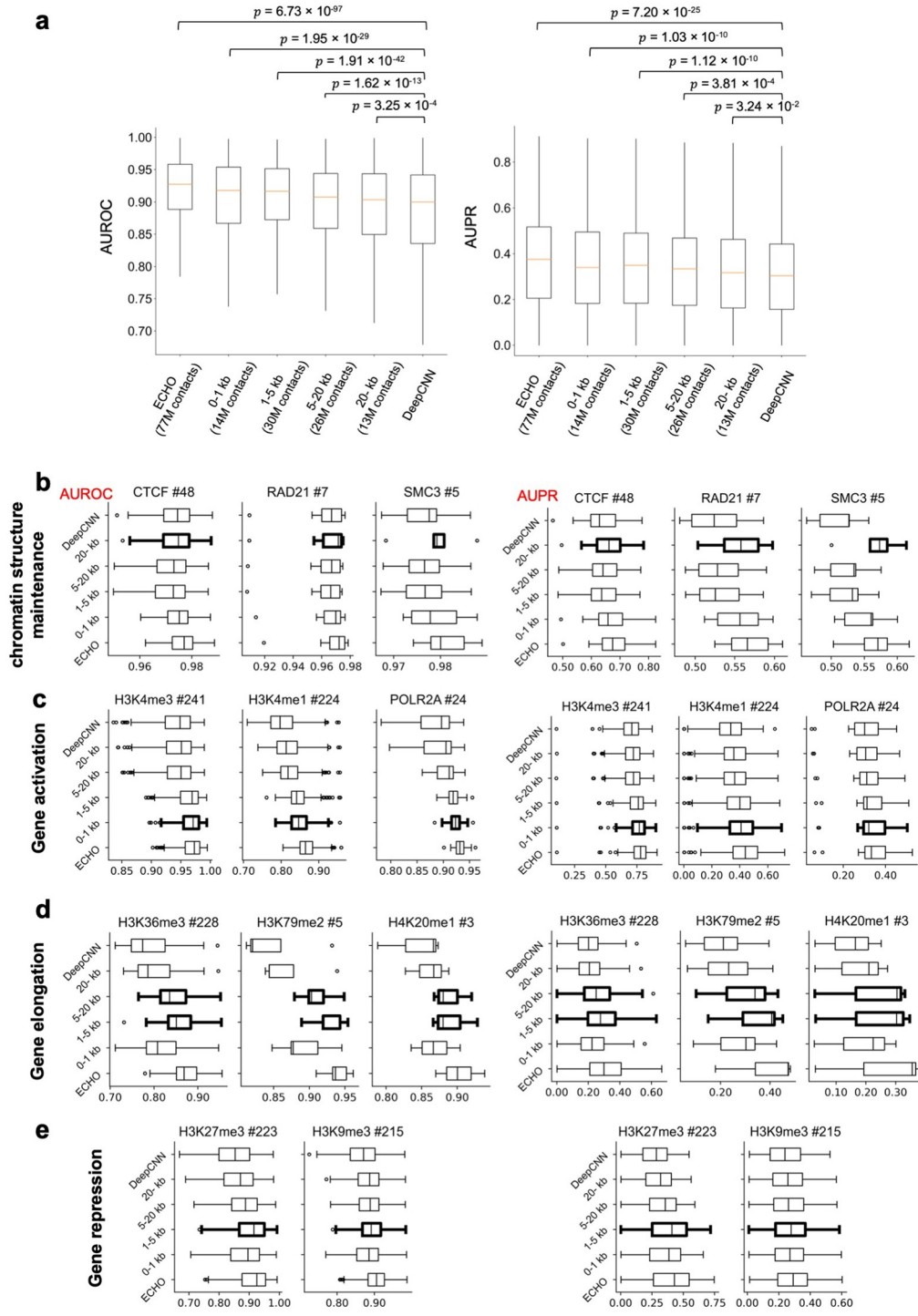

**Fig 2. How much chromatin contacts helps to predict a chromatin feature is contact-distance specific.** (a) Comparing the performance of ECHO using Micro-C contacts from specified contact distance ranges. The x-axis shows four groups with different contact distance ranges and the total number of Micro-C contacts used. The *p*-values calculated by paired *t*-tests are given. (b-e) show the effects of Micro-C contact distances on the prediction of chromatin structure maintenance related TFs, gene activation related chromatin features, gene elongation marks, and gene repressive marks, respectively.

## Chromatin contact resolution is critical in ECHO's prediction of chromatin features

To investigate whether the resolution of chromatin contacts is critical to improve the prediction performance, we compared the performance of ECHO using the high-resolution Micro-C contact maps and the relatively low-resolution Hi-C contact maps. In the previous settings, high-resolution 200-bp Micro-C contact maps were used to provide spatial neighborhood information for ECHO. Different from Hi-C which used a restriction enzyme [19], Micro-C used MNase for chromatin fragmentation [20], providing higher resolution than Hi-C. Since Hi-C could not reach the resolution higher than 1 kb, we sampled 200bp chromatin contacts from the 1 kb resolution Hi-C contact maps as follows. For each 200bp central sequence, we first found the 1kb bin where it located so that we could identify its 1kb spatial neighbors from 1kb Hi-C contact maps. Then, for each of these 1kb spatial neighbors, we connected the centered 200bp bin with the original 200bp central sequence to form a 200-bp resolution Hi-C contact map. The motivation of this sampling strategy was that the centered 200bp bin with the flanking regions could also cover the whole 1kb region. Consistent with the normalization of Micro-C contact maps, both the HFF and H1-hESC Hi-C contact maps were normalized and merged by taking the maximum contact value at each position to generate a weighted adjacency matrix, and contacts with low contact values were filtered out with some threshold. In this experiment, we chose two thresholds and sampled 50 spatial neighbors and 10 sequential neighbors for each sequence similarly, resulting in 61M and 83M Hi-C contacts, respectively. ECHO using Micro-C contact maps predicted chromatin features with a mean AUROC of 0.921, which was significantly higher than ECHO with 83M Hi-C contacts (AUROC 0.914, $p$-value 7.08E−6), ECHO with 61M Hi-C contacts (AUROC 0.912, $p$-value 4.67E−10) and DeepCNN (AUROC 0.885, $p$-value 6.73E−97). Moreover, ECHO with Micro-C contact maps yielded a mean AUPR of 0.378, which outperformed ECHO with 83M Hi-C contacts (AUPR 0.367, $p$-value 4.92E−2), ECHO with 61M Hi-C contacts (AUPR 0.361, $p$-value 4.79E−3) and DeepCNN (AUPR 0.318, $p$-value 7.20E−25). As the results demonstrated, ECHO with either Micro-C or Hi-C contact maps achieved better performance than baseline DeepCNN. However, ECHO with Micro-C contact maps using fewer contacts outperformed ECHO with Hi-C contact maps, which indicated that high-resolution Micro-C contact maps provided more precise chromatin contact information for predicting chromatin features (Fig 3).

## Identifying important chromatin contacts contributing to predicting chromatin features

For characterizing the contributions of Micro-C contacts and DNA sequences towards predicting chromatin features, we applied an attribution method [21] to our ECHO framework, which calculated the attribution scores of the inputs as *gradient × input* (Fig 4a and Methods). Unlike sequence-based models which only attributed the chromatin features to the sequence itself, ECHO also attributed chromatin feature prediction to the sequences of its neighbors implied by chromatin contacts, which characterizes the contributions of neighbor sequences to the chromatin features.

Moreover, the attribution scores computed on the contact matrices identified the important contacts relevant to the targeted chromatin feature (S8 Fig). To validate the contact attribution, we took CTCF for example. As an architectural protein bridging genome topology and functions [22], CTCF interactions play an important role in DNA looping and transcriptional regulation. Suppose that each contact connected one central sequence and one of its neighbor sequences, we first identified all chromatin contacts whose central sequences were bound to CTCF, and calculated their attribution scores related to CTCF prediction. Then the contacts

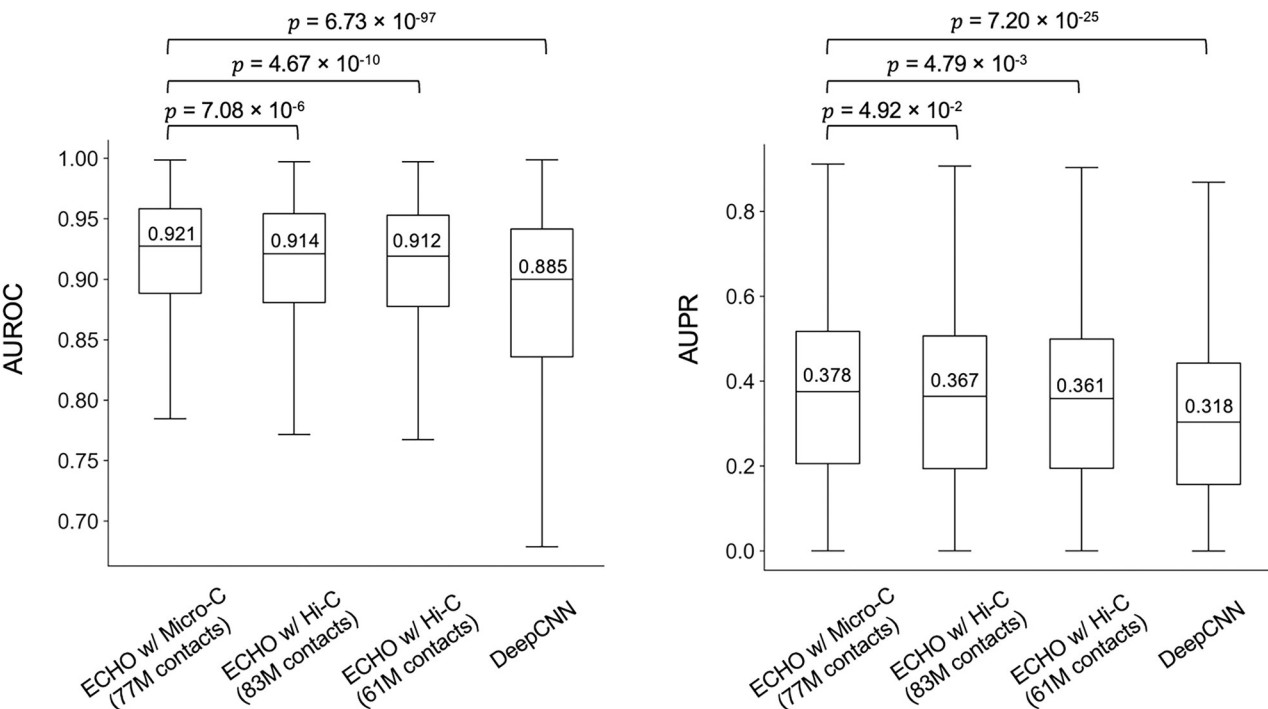

**Fig 3. ECHO's performance when it is coupled with different types of contact maps.** Two types of contact maps (200-bp Micro-C contact maps and up-sampled 200-bp Hi-C contact maps) are compared here. The *p*-values from paired *t*-tests indicate that Micro-C contact maps provide more precise high resolution chromatin contact information for chromatin feature prediction. The x-axis shows the contact map types and the total numbers of chromatin contacts used by ECHO (resulting from two contact value cut-off thresholds). The median AUROC and AUPR scores are provided above the median lines.

with attribution scores greater than 0.1 were divided into four groups with contact distance between 5–10 kb, 10–30 kb, 30–100 kb, and over 100 kb. In each group, we calculated the proportion of these contacts whose other anchor was also bound by CTCF, namely the percentage of CTCF interaction (Fig 4b and 4c).

Additionally, we added a baseline which identified all Micro-C contacts whose central sequence bound by CTCF without filtering the attribution scores, and obtained the percentages of interactions whose both anchors were bound by CTCF, which were shown as 'all' in Fig 4c. We observed that the percentage of potential CTCF interactions increased with the chromatin contact distance. However, the increasing rate was significantly lower than the increasing rates of the percentages of CTCF interactions stratified by attribution scores, which could be observed from the other four groups filtered by attribution scores in the same figure. Within short-range interactions, the percentages of CTCF interactions were significantly lower than those within long-range interactions. However, as the contact distance increased, long-range CTCF interactions were more essential to CTCF binding prediction. When predicting the binding sites of CTCF, if ECHO assigned a high attribution score to a >30 kb Micro-C contact, then CTCF also bound the other anchor of the contact with a very high probability (Fig 4c). Moreover, this pattern still existed when we changed the threshold of attributions scores to 0.3, 0.5, and 0.7. The distance range matched well with the common CTCF-mediated chromatin looping, indicating that ECHO not only captured the relationships between loops and CTCF interactions but also enhanced the prediction of CTCF binding sites by identifying its potential interacting domains.

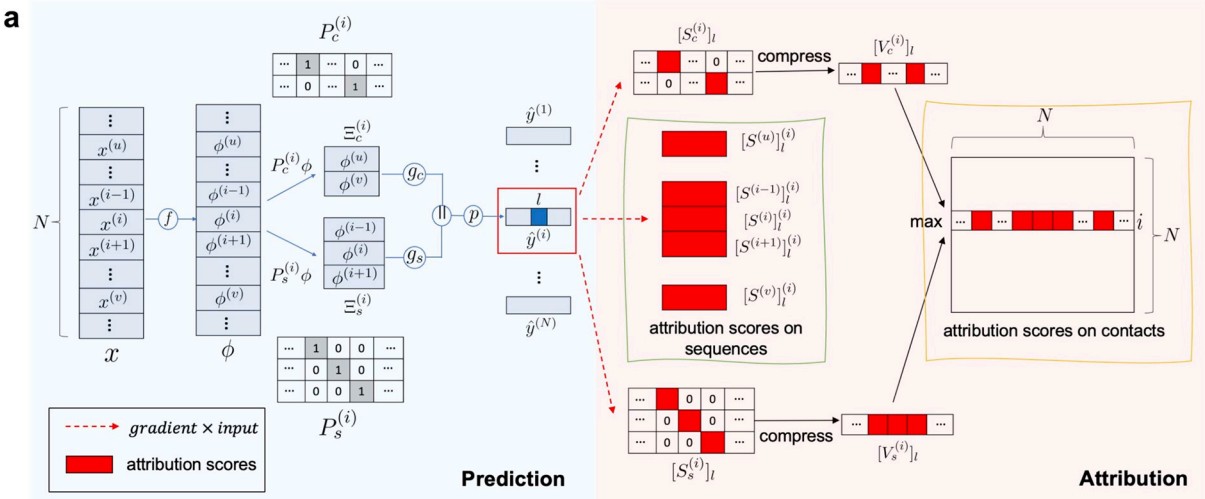

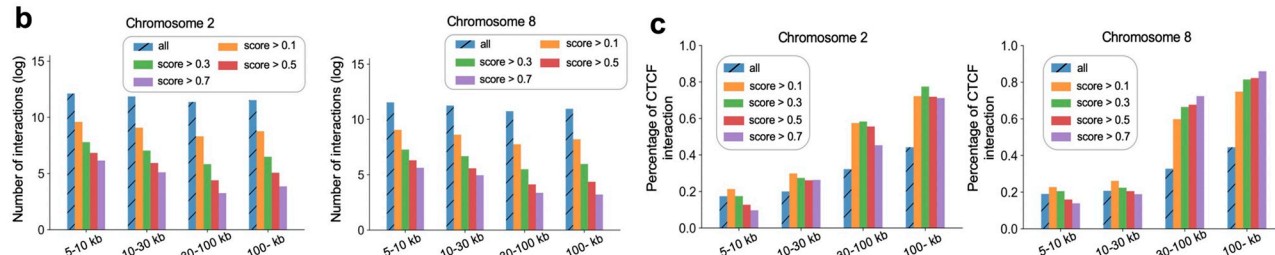

**Fig 4. Details of attribution methods and validations of attributions on Micro-C contacts.** (a) Workflow of attribution methods. If sequences *u* and *v* are the spatial neighbors of the central sequence *i*, sequences *i*−1 and *i* + 1 are *i*'s sequential neighbors, then in the forward propagation, sequence layers *f* are applied to extract hidden representation $\phi^{(i)}$ for each sequence. Next, two binary sampling matrices, $P_c^{(i)}$ and $P_s^{(i)}$, are input to multiply with $\phi$ to form two feature matrices, $\Xi_c^{(i)}$ and $\Xi_s^{(i)}$. The two feature matrices are fed into graph layers ($g_c$ and $g_s$) and one prediction layer *p* to compute a predicted chromatin feature vector $\hat{y}^{(i)}$. Next we take the gradient of the specific label *l* on *i* with respect to the inputs (two sampling matrices, the central sequence and neighbor sequences) independently, the attribution scores of each input which indicate the contributions to the central sequence *i*'s label *l*, are calculated by *gradient × input* [21]. The attributions on the two sampling matrices, $[S_c^{(i)}]_l$ and $[S_s^{(i)}]_l$, are first compressed into two vectors, $[V_c^{(i)}]_l$ and $[V_s^{(i)}]_l$, and then taken the maximum to become the *i*-th row in the interaction importance matrix. In addition, the attribution scores $[S^{(i-1)}]_l^{(i)}$ of the sequence *i*−1 for label *l* on sequence *i* are computed. (b) Logarithm plots to show the total numbers of Micro-C contacts whose central sequence side bound by CTCF. Micro-C contacts are separated into four groups according to the contact distances: 5−10 kb, 10−30 kb, 30−100 kb, and 100− kb. In each group, the contacts with CTCF related attribution scores greater than 0.1, 0.3, 0.5, and 0.7 are selected, and group 'all' is a baseline where contacts are not filtered by attribution scores. (c) Bar-plots to show the percentage of CTCF interactions, which is calculated as the number of contacts whose two sides were bound by CTCF over the total number of contacts whose central sequence was bound by CTCF.

Furthermore, to test whether the attribution scores related with chromatin features could reflect cell-type specific Micro-C patterns, we calculated the attribution scores on the chromatin contacts related with the chromatin features in H1 cell line whose Micro-C contact map existed, and compared them with the contact map used by ECHO. Since not all of the chromatin contacts were related to chromatin features to be predicted by ECHO, we did not expect to recover the comprehensive Micro-C patterns from the attribution scores. The attribution scores showed more similarity with the chromatin contact maps in some regions than the others. Here, we gave two example regions in S9 Fig. In one genomic region, the attribution scores resembled the original chromatin contact matrices, but in the other region, the attribution scores and the contact matrices showed less similarity.

## Identifying TF binding motifs and collaborative binding mechanisms among TFs

Previous chromatin feature prediction models like Enformer [23] and Basenji [11] did not focus on capturing TF collaborative binding. Although collaborative binding mechanisms could be potentially captured by the attention strategy used by Enformer and the dilated convolution strategy used by Basenji, the collaborative binding activities that could be potentially recovered by these models were limited within their receptive fields (e.g., up to 100kb in Enformer). Another prediction model SATORI could capture collaborative binding but it was limited to a small region (within 1kb distance). By contrast, ECHO leveraged Micro-C contact maps and could identify collaborative bindings guided by chromatin contact maps, without a range limit. The collaborative binding mechanisms were reflected by the attribution scores calculated on both the central sequence and its neighbor sequences related to chromatin features on the central sequence. We first found that the highly attributed regions on the central sequences matched known binding motifs from JASPAR [24] (S10 Fig). Furthermore, some high attribution score regions in the neighbor sequences which contributed to chromatin feature prediction on the central sequence, also matched known motifs from JASPAR. Therefore, the correlated high attribution score regions on both central and neighbor sequences might reflect TF collaborative binding (S10 Fig, and the first example showing collaborative binding patterns with a distance of 190kb). In addition, TF binding motifs can be identified from attribution scores of corresponding TF binding sequences by using TF-MoDISco [25] (S10 Fig).

The TF collaborative binding mechanisms were further investigated by combining attributions on DNA sequences and Micro-C contacts. For example, a contact between two candidate cis-regulatory elements (cCREs) from ENCODE [26] at chr 2: 28,810,975−28,811,136 and chr 2: 28,821,422−28,821,694 received an attribution score of 1.0 for CTCF labels. By attributing the TFs binding on the central sequence to both the cCREs, we found that the high attribution score regions on the two cCREs matched with the CTCF motif from JASPAR (Fig 5). This indicated that ECHO learned both the CTCF motifs and the CTCF interaction patterns, and the CTCF on neighbor sequence's side cCRE contributed to the binding of CTCF on the central sequence's side cCRE.

We also found a candidate enhancer-promoter interaction between chr 2: 113,627,160 −113,627,340 and chr 2: 113,602,125−113,602,462 receiving an attribution score 0.925 related with CTCF binding (Fig 5b). By calculating the attribution scores for TFs binding on the central sequence, we identified an ELF1 motif on the central sequence side's cCRE and three TF binding motifs on the neighbor sequence side's cCRE, which contributed to TFs binding on the central sequence. First, we identified the binding motif of ZBTB3 which frequently appears in the proximity to CTCF [27], from the neighbor sequence's attribution scores, but we found that ZBTB3 was not selected in our TF profiles (i.e., ZBTB3 was not predicted). Since sequence-based models such as DeepCNN could only attribute the predicted binding TFs to its central sequence, this motif might not be detected from DeepCNN's attribution scores. To test if DeepCNN could detect this motif, we directly attributed the sequence which contained the ZBTB3 motif based on its predicted binding TFs by using DeepCNN, and the ZBTB3 binding motif was not detected from the attribution scores as expected. Therefore, DeepCNN's attribution scores could miss some binding motifs of unpredicted TFs, but ECHO might detect these binding motifs in the neighbor sequences by identifying important regions which contributed to TF binding prediction on the central sequence. Furthermore, the NRF1 motifs identified by ECHO and DeepCNN were not in the same location, but the attribution scores assigned by ECHO matched the NRF1 motif better than DeepCNN (see Fig 5b and the high

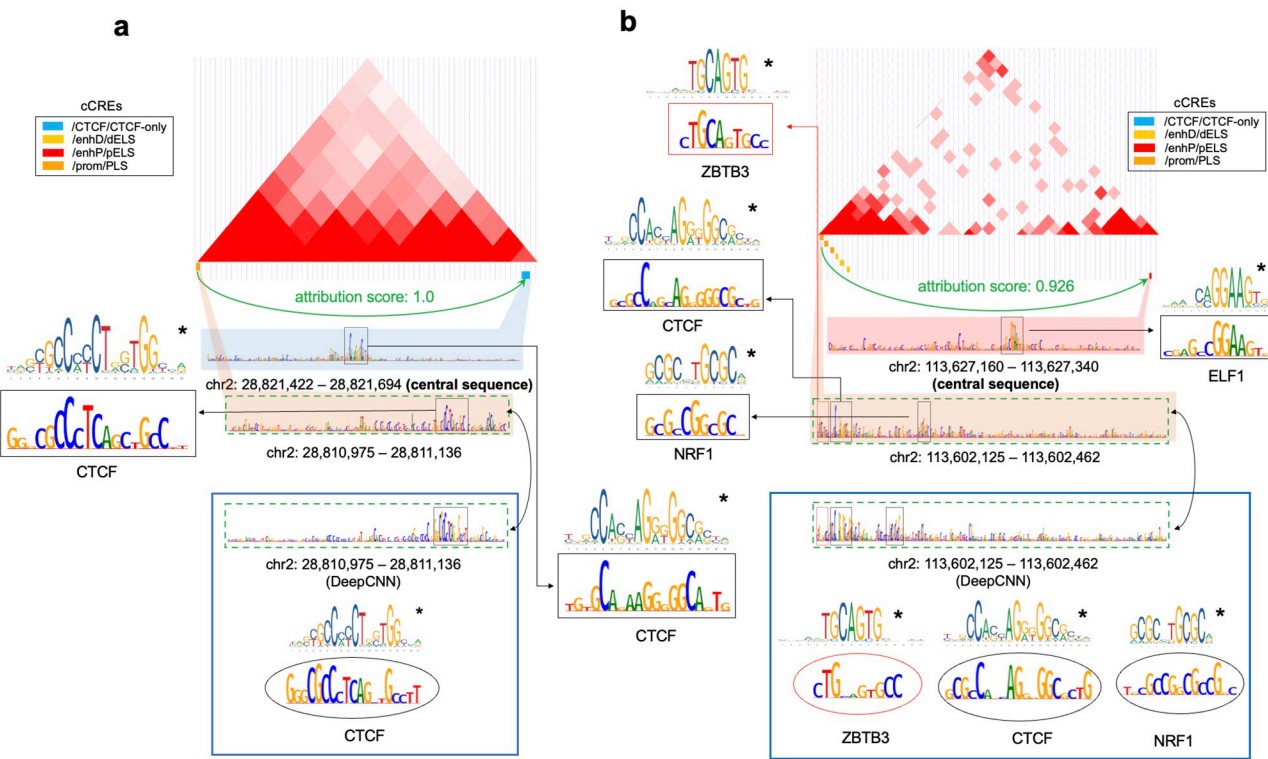

**Fig 5. Visualization of attribution scores on chromatin contacts, central sequences, and neighbor sequences.** The hESC Micro-C contact map is visualized in the UCSC genome browser [28]. The green arrow represents a Mirco-C contact pointing to the central sequence side. The attribution scores of the Micro-C contacts for CTCF are provided. The candidate cis-regulatory elements (cCREs) within both anchors of the contact are shown as small colored rectangles, which are attributed for TFs binding on the central sequence. High attribution score regions are plotted in the black rectangles. The binding motifs from JASPAR are marked with ⋆. As a comparison, the attribution scores of cCREs on the neighbor sequence sides contributing to TFs binding on the neighbor sequences are computed using the baseline DeepCNN, which are shown in the large blue blocks. The corresponding high attribution score regions are plotted in ovals. (a) The high attribution score regions on both the two cCREs match known CTCF motifs from JASPAR. (b) The regions within the red rectangles match binding motifs of TFs not included in our collected data. The attribution scores of the neighbor sequence's cCREs computed by ECHO identify ZBTB3 which was not identified by DeepCNN. The attribution scores from ECHO detected NRF1's motif, which is more proximate than the one identified by DeepCNN.

attribution score regions highlighted in rectangles for ECHO and highlighted in ovals for DeepCNN).

### Discovering sequence patterns in the neighborhood contributing to chromatin features on the central sequences

In this experiment, we investigated whether specific sequence patterns in the spatial neighborhood existed and contributed to the related chromatin features on the central sequences. From the previous results, ECHO outperformed sequence-based models on multiple individual chromatin feature prediction (Fig 1c). Sequence-based models performed worse, particularly on TFs without known motifs, since they failed to extract patterns from the central sequences. However, ECHO, by leveraging neighborhood information, improved the prediction accuracy more significantly on TFs without known motifs than TFs with known motifs (Fig 6c, S11 and S12 Figs). Thus, we believed that ECHO's improvement in chromatin feature prediction largely came from leveraging neighbor sequences.

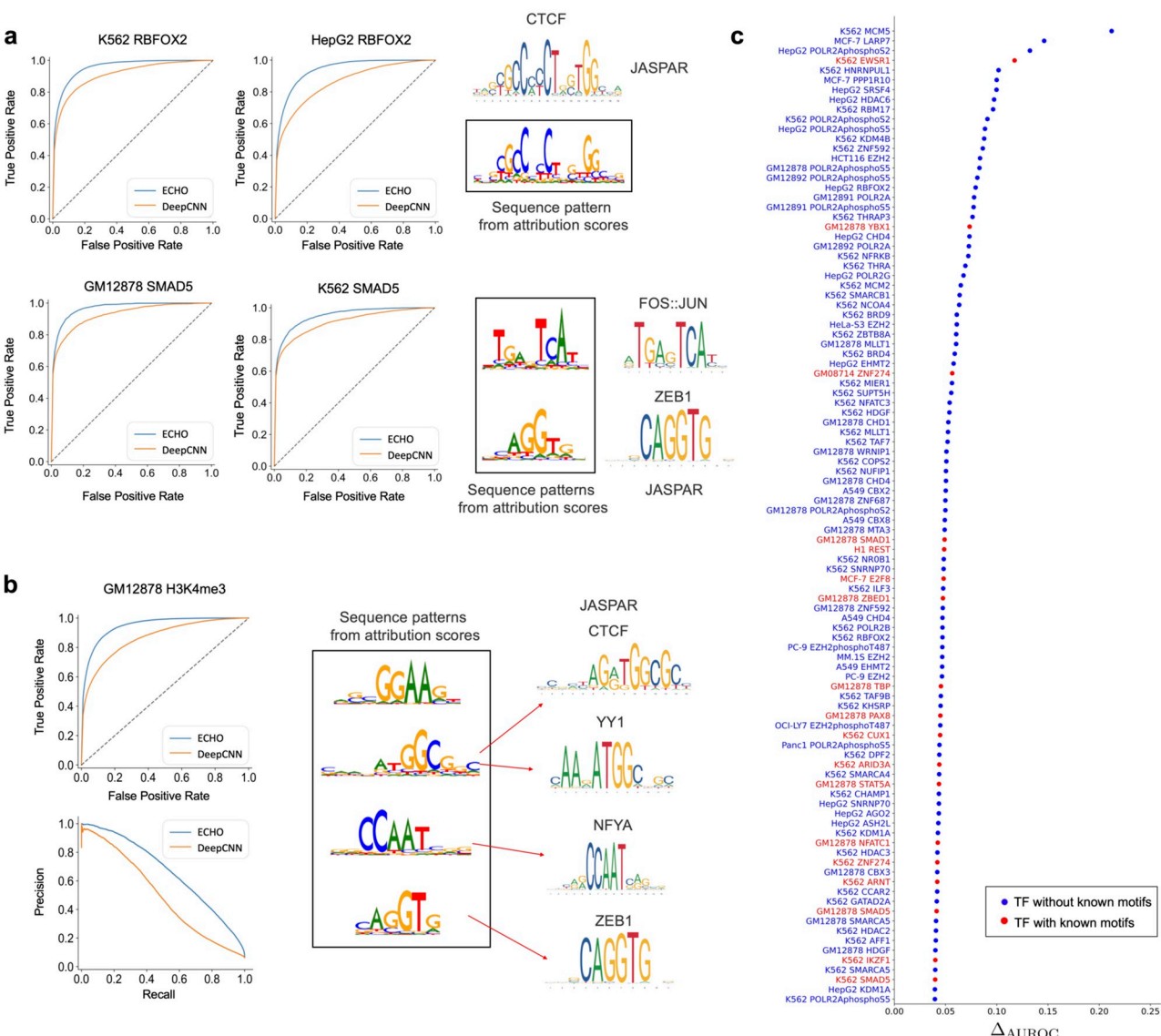

**Fig 6. Model performance improvement compared to DeepCNN and visualization of sequence patterns in the neighborhood of investigated chromatin features.** (a) ECHO predicts RBFOX2 and SMAD5 more accurately on all the collected cell lines than the baseline DeepCNN. The patterns generated by TF-MoDISco [25] using the selected neighbor sequences' attribution scores for the corresponding TFs binding on central sequences, which are compared with known binding motifs from JASPAR [24]. (b) ECHO outperforms DeepCNN on GM12878 H3K4me3 prediction and the important patterns in the neighborhood are visualized. 400 neighbor sequences which have high attribution score contacts with successfully predicted GM12878 H3K4me3 sites are selected. The attribution scores of the neighbor sequences are input to TF-MoDISco to identify frequent patterns. Top four patterns are given, and some of them match several TFs' binding motifs from JASPAR. (c) 100 TFs with the cell lines which achieve greatest improvement on AUROC compared to DeepCNN.

This motivated us to capture specific patterns contained in the neighbor sequences and helpful to TF binding. We first calculated attribution scores of chromatin contacts for investigated TFs. Then we selected the highly attributed contacts whose central sequence sides were these TFs' binding sites, and these binding events could be successfully predicted by ECHO. Next, two hundred neighbor sequences from the other anchor of these chromatin contacts were sampled. Next, common sequence patterns were identified from the attribution scores with respect to TFs binding on the central sequences by TF-MoDISco [25].

Examples were given to validate our assumptions (Fig 6a). We first found that ECHO predicted SMAD5 more precisely on all the collected cell lines. Then a FOS:JUN related pattern and a CAGGTG pattern were identified in its neighborhood. Corresponding evidences were found that SMADs had a low affinity for DNA, so they relied heavily on co-factors for the transcriptional regulation [29]. For ZEB1 whose binding motif contains CAGGTG pattern, it bound to coactivator p300 and was highly correlated with SMAD binding by promoting the formation of a p300–Smad transcriptional complex [30]. Additional evidence was found that SMAD interacted with AP-1 [31] whose binding site was identified as 5'-TGA G/C TCA-3', and FOS:JUN is a subunit of AP-1.

We further explore whether these significant patterns existed around histone marks. We identified sequence patterns regarding H3K4me3 on cell line GM12878. For all the central sequences predicted to be H3K4me3 in GM12878, we identified four hundred neighbor sequences from important contacts, and their attribution scores were studied by TF-MoDISco. In the end, important patterns were identified and validated by current biological knowledge (Fig 6b). One of the pattern, ATGGCGG, matched parts of CTCF and YY1 motifs, and both YY1-binding and CTCF-YY1 co-binding regions were known to be enriched by H3K4me3 [32]. Another pattern matched NFYA motif from JASPAR, which agreed with the fact that the presence of H3K4me3 depended on NF-Y [33].

## Discussion

Our model predicts chromatin features from DNA sequences in the light of high-resolution 3D chromatin organization, which differs from previous sequence-based deep learning models which only utilize DNA sequences. As a result, ECHO significantly outperforms sequence-based models on chromatin feature prediction in terms of AUROC and AUPR scores. The improvement is consistent in terms of the specific sequence-based model ECHO builds on and the specific chromatin feature to be predicted. ECHO also outperforms the only graph-based chromatin feature prediction model in the literature, ChromeGCN [34], which demonstrates that convolution over sampled neighborhood strategy in ECHO is important.

Whereas ECHO and ChromeGCN [13] explicitly leverage chromatin contacts, DNA interactions are implicitly captured by SATORI [12], Basenji [11], Enformer [23]. SATORI captures TF-TF interactions by combining CNN with self-attention mechanisms. SATORI first finds interactions between positions on DNA sequences with attention matrices, then position interactions are converted to CNN filter interactions and finally translated to TF interactions. However, SATORI captures interactions within the short input DNA sequence and cannot capture long-range interactions. Basenji and Enformer allow long DNA sequences as input to capture long-range interactions and predict signal track data. Basenji adopts dilated convolution to increase receptive field size which enables to deal with 131 kb long DNA sequences. Enformer uses self-attention to capture long-range interactions and allows longer input sequences than Basenji. Unlike Enformer and Basenji, ChromeGCN and ECHO leverage 3D chromatin organization which aggregate the regulatory information from the chromatin spatial neighbors without a range limit, so that ChromeGCN and ECHO identify important chromatin contacts contributing to chromatin feature prediction by attributing these chromatin contacts. Furthermore, ECHO identifies collaborative bindings by discovering sequence patterns on both the central sequences and the spatial neighbor sequences, while capturing collaborative bindings is not a focus of Enformer and Basenji. Moreover, ECHO uses Micro-C rather than Hi-C used by ChromeGCN, which better captures fine-scale chromatin contacts and is more desirable to capture interactions between chromatin features. The methodologies used in ChromeGCN and ECHO are also quite different, ChromeGCN converts the entire DNA sequence into the

graph structure by leveraging Hi-C contact maps, and uses a gated graph convolution network (GCN) to aggregate the neighborhood information while the gated function determines whether to use the aggregated neighborhood information. Nonetheless, ECHO further transforms the graph structure data from higher-resolution contact maps to grid structure and applies 1D convolutions to learn the neighborhood aggregation patterns.

Our work also systematically characterizes the contributions of the spatial and sequential genomic neighborhood towards predicting the chromatin features of a central sequence. Different from previous sequence-based prediction models which only calculate attribution scores on each central sequence for its own chromatin features, and also different from ChromeGCN which only computes the attribution scores on the Hi-C contacts excluding DNA sequences, ECHO attributes the chromatin feature prediction to both chromatin contacts and DNA sequences (including central sequences and their neighbors). Moreover, ECHO, which mainly consists of convolutional layers, is more interpretable compared to ChromeGCN. Some popular interpretation methods, such as gradient-based attribution methods and visualizing first convolutional layers, may not perform well on ChromeGCN for the following reasons. First, the full batch training in ChromeGCN is computationally prohibitive to compute the gradients end-to-end. Other typical gradient-based attribution methods, such as DeepLIFT [21] and layer-wise relevance propagation (LRP) [35], may not perform well, since original DeepLIFT cannot handle gating functions in ChromeGCN and LRP can fail with Sigmoid activation function [34]. In addition, the motifs learned from the first convolutional layer are the same as motifs learned from sequence-based models, since pre-training sequence-based models freezes the first convolutional layer in ChromeGCN.

Another contribution in our work is that ECHO with neighborhood sampling enables to deal with large-scale graphs and datasets [16]. The data used by ECHO includes 200-bp high-resolution Micro-C contact maps with more than 77 million contacts, and more than 2.9 million input DNA sequences, and 2,583 chromatin features.

One of the limitations in our work is that some valuable chromatin contacts may be overlooked in ECHO's neighborhood sampling procedures. For each central sequence, ECHO samples a number of neighbor sequences according to the contact strength. When one central sequence has multiple neighbor sequences, some important Micro-C contacts with weaker contact strength may be left out. Another limitation is that we only merge two Micro-C contact maps (hESC and HFF) to provide chromatin contact information, but the chromatin features are from multiple cell lines. Although many chromatin structures are preserved across different cell types, we suspect some of the chromatin contacts related to transcription regulation are cell type-specific. Therefore, we anticipate ECHO performs better when additional Micro-C data sets become available in the future.

## Methods

Different from previous chromatin feature prediction models, including DeepSEA [7], DanQ [8], DeepBind [9], and Basenji [11], which ignore the important 3D chromatin organization, our model ECHO extracts both sequential features along the DNA sequence and spatial features from $\mathcal{G}$, which explicitly describes high-resolution 3D chromatin organization. In the graph $\mathcal{G} = (\mathcal{V}, \mathcal{E}, \mathcal{A})$, each node $v \in \mathcal{V}$ represents a non-overlapping DNA segment of fixed length (i.e., 200 bp), and $\mathcal{V}$ covers the entire DNA sequence. Each edge $e \in \mathcal{E}$ represents a chromatin contact between two DNA segments, and the strength of the chromatin contacts is captured by $\mathcal{A}$. Because some of the 3D chromatin contacts are cell-type specific, we merge several existing Micro-C contact matrices from different cell types by taking the maximum contact value at each entry of the adjacency matrix $\mathcal{A}$.

## Model description

ECHO takes a one-hot encoding representation of one DNA sequence segment $x^{(i)}$ as input, and computes a vector of chromatin features $\hat{y}^{(i)}$ with a number of neural networks layers, including sequence layers, graph layers, and one prediction layer. Since each layer in a deep learning model is a function mapping the layer inputs to the layer outputs, we use functions $f$, $g$, and $p$ to represent sequence layers, graph layers, and prediction layer, respectively. The architectural and algorithmic details of our model are given in Fig 1a and Algorithm 1, and the notations used in our work are summarized in S1 Table. Next, we describe each of the three layers in detail.

The sequence layers $f$ are designed to extract sequence feature information

$$\phi^{(i)} = f(x^{(i)}) \tag{1}$$

where $i \in \{1, \ldots, N\}$ is the index of 1000-bp DNA sequences (including 400-bp flanking regions both upstream and downstream a 200-bp central sequence) ordered by their locations on the reference genome, and $N$ indicates the number of input sequences. $\phi^{(i)} \in R^K$ represents the hidden representation of sequence $i$ from the sequence layers $f$. To reduce training parameters and accelerate training, we adopt a pre-train approach which is also used in ChromeGCN [13]. Existing sequence-based models, e.g. DanQ [8] and DeepSEA [7] are first pre-trained to learn features from sequences, and all the pre-trained layers except the last fully connected layer in the sequence-based models are frozen as our sequence layers $f$.

Since the convolution kernels operate on receptive fields of fixed size and order, original convolutions cannot be performed on a graph [14]. Therefore, between the sequence layers and graph layers, functions SAMPLE and STACK are introduced to transform the graph structure data into grid structure data. Specifically, we first sample a fixed number of sequences from the neighbor set and stack the hidden representations of these sequences to form a feature matrix $\phi$ which allows convolution operations to extract sequential and spatial features, namely

$$\Xi_s^{(i)} = \text{STACK}\left(\text{SAMPLE}(\phi, \mathcal{N}_s^{(i)})\right), \tag{2}$$

$$\Xi_c^{(i)} = \text{STACK}\left(\text{SAMPLE}(\phi, \mathcal{N}_c^{(i)})\right). \tag{3}$$

Let $\mathcal{N}_s^{(i)}$ represent a sequential neighbor set containing the neighbors along the chromatin fiber. Let $\mathcal{N}_c^{(i)}$ indicate a spatial neighbor set from the 3D chromatin structure, i.e., a union of sequences which contact with central sequence $i$, and here only the first-order neighbors with direct contacts are considered. The neighbor set is defined as the union of these two sets, namely

$$\mathcal{N}^{(i)} = \mathcal{N}_s^{(i)} \cup \mathcal{N}_c^{(i)}. \tag{4}$$

Suppose that $k_s$ sequential neighbors from both downstream and upstream of the central sequence $i$ are sampled, then we stack the hidden representations of these selected sequences

according to their location orders on the genome to generate a feature matrix

$$
\Xi_s^{(i)} = \begin{bmatrix} \left(\phi^{(i-k_s)}\right)^T \\ \left(\phi^{(i-k_s+1)}\right)^T \\ \vdots \\ \left(\phi^{(i+k_s)}\right)^T \end{bmatrix} \in R^{(1+2k_s)\times K}. \tag{5}
$$

Next the spatial neighbor set is defined as

$$
\mathcal{N}_c^{(i)} = \{t | \mathcal{A}[i,t] > \tau\} \tag{6}
$$

where $\mathcal{A}[i,t]$ is the normalized contact value between sequences $i$ and $t$, and $\tau > 0$ is a threshold to filter out extremely small contact values and noisy contacts. Then we order the sequences from the neighbor set $\mathcal{N}_c^{(i)}$ by the contact values with the central sequence $i$, then $k_c$ sequences with top contact values are sampled from the set $\mathcal{N}_c^{(i)}$. Similarly, a feature matrix $\Xi_c^{(i)} \in R^{k_c \times K}$ is generated by stacking the hidden representations of sampled sequences with an order of the contact values. If one central sequence does not have as many neighbors as expected, dummy sequences $M' \in R^K$ with features of all zeros are added to feature matrices to ensure they are of the same size. Although the sequential neighbors are also likely to be spatial neighbors from Micro-C contact maps, independently sampling the sequential neighbors is able to inform the model that these neighbor sequences are the nearest potential TF binding sites to the central sequence, which may affect the central sequences' chromatin features along the DNA sequence.

The graph layers $g$ perform convolution on the feature matrices $\Xi_c^{(i)}$ and $\Xi_s^{(i)}$ of central sequence $i$ as well as its neighbors to aggregate information from the neighborhood,

$$
h_s^{(i)} = g_s(\Xi_s^{(i)}), \quad h_c^{(i)} = g_c(\Xi_c^{(i)}), \tag{7}
$$

where $h_s^{(i)}$ and $h_c^{(i)}$ are updated hidden representation extracted by graph layers $g_s$ and $g_c$. The feature matrix $\Xi_c^{(i)} \in R^{k_c \times K}$ enables 1D convolution by taking $k_c$ as the number of channels and $K$ as the feature size, so as the feature matrix $\Xi_s^{(i)}$. For the two types of feature matrices, their corresponding graph layers $g_c$ and $g_s$ are applied. The structures of graph layers $g_c$ and $g_s$ are the same except for the first layer which receives inputs in different sizes (the numbers of sampled sequential and spatial neighbors are different). Both $g_c$ and $g_s$ consist of convolution layers and a global average pooling [36] in the last layer. Each final feature map reflects one type of sequential and spatial neighborhood information aggregation pattern.

The prediction layer $p$ predicts chromatin features from the updated hidden representations of each input sequence,

$$
\hat{y}^{(i)} = p(h_s^{(i)} \| h_c^{(i)}), \tag{8}
$$

where $\|$ indicates a concatenation operation. The hidden features output by the graph layers $h_s^{(i)}$ and $h_c^{(i)}$ are concatenated as the embedding of the central sequence $i$, which is fed into the prediction layer $p$ with one fully connected layer.

Although higher-order neighborhood information is not investigated in our work, it can also be learned by mixing powers of adjacency matrix [37]. For example, to learn the neighborhood information within an $n$-th order, additional spatial neighbor lists $[\mathcal{N}_c^{(i)}]^j$ need to be sampled from the $j$-th power of adjacency matrix $\mathcal{A}^j$ to generate the feature matrix $[\Xi]^j$. Then a

new embedding $\|_{j=2,\ldots,n} g_c^j([\Xi^{(i)}]^j)$ is concatenated to the first-order node embedding where $g_c^j$ are the graph layers corresponding to $j$-th order neighborhood information extraction.

**Algorithm 1:** workflow of ECHO

```
Input: Graph (𝒱,ℰ,𝒜). Input features x⁽ⁱ⁾ ∀i = 1,⋯,|𝒱|. Sequence layers f.
Graph layers g_c and g_s. Prediction layers p. Number of sampled sequence
and first-order neighbors k_s and k_c. First-order neighbor list
𝒩_c⁽ⁱ⁾ ∀i = 1,…,|𝒱|.
Output: Predicted chromatin features ŷ⁽ⁱ⁾  ∀i = 1,…,|𝒱|.
```

$\phi^{(i)} \leftarrow f(x^{(i)}) \quad \forall i = 1,\cdots,|\mathcal{V}|$

**for** $i = 1,\ldots,|\mathcal{V}|$ **do**

 Dummy node $M' \leftarrow \mathbf{0}$

 $\Xi_s^{(i)} \leftarrow [\phi^{(i-k_s)},\cdots,\phi^{(i+k_s)}]^T$

 $\{i^{[1]}, i^{[2]},\cdots, i^{[\|\mathcal{N}_c^{(i)}\|]}\} \leftarrow \text{sort}(\mathcal{N}_c^{(i)})$

 **if** $|\mathcal{N}_c^{(i)}| < k_c$ **then**

 $\Xi_c^{(i)} \leftarrow [\phi^{(i^{[1]})},\cdots,\phi^{(i^{[\|\mathcal{N}_c^{(i)}\|]})},\cdots,M']^T$

 **else**

 $\Xi_c^{(i)} \leftarrow [\phi^{(i^{[1]})},\cdots,\phi^{(i^{[k_c]})}]^T$

 **end**

**end**

$h_s^{(i)} \leftarrow g_s(\Xi_s^{(i)}), \quad h_c^{(i)} \leftarrow g_c(\Xi_c^{(i)}) \quad \forall i = 1,\cdots,|\mathcal{V}|$

$\hat{y}^{(i)} \leftarrow p(h_s^{(i)}\|h_c^{(i)}) \quad \forall i = 1,\cdots,|\mathcal{V}|$

## Model training

For training our models, we first downloaded the human reference genome GRCh38 and removed sequence gaps and unannotated regions. The rest of the genome was segmented into 200-bp bins. Next we collected 2,583 chromatin feature profiles including 882 TFs, 1,510 histone marks and 191 DHS profiles from the Encyclopedia of DNA elements (ENCODE) [26] and the International Human Epigenome Consortium (IHEC) [38]. The bins were labelled in a way that if more than half of a bin was in the peak region, then the corresponding entry in its chromatin feature vector was set to be 1, and 0 otherwise. We had over 2.9 million 200-bp segments bound with at least one TF, resulting in 585, 137,600-bp DNA sequence (20.4% of the human reference genome). We added a 400-bp flanking region to both upstream and downstream of the 200-bp sequence to generate a 1000-bp sequence. As input, each sequence was represented by a $4 \times 1000$ one-hot encoding matrix. These input sequences were split to a training set, a validation set, and a testing set without overlapping. Here we chose chromosomes 2, 8, and 21 as the testing set, chromosomes 3 and 12 as our validation set, and the remaining chromosomes were used to train our model.

The default pre-train model in ECHO was DeepCNN if not specified. The length of each hidden feature vector $\phi^{(i)}$ extracted by sequence layers $f$ was 2600. The weighted adjacency matrix $\mathcal{A}$ was generated by merging Micro-C contact matrices of two cell lines, HFF and hESC as follows. We first calculated the total read counts in both the contact maps for each chromosome and the ratio between the two maps. Each contact value in the contact matrix with a larger total read count was multiplied with the ratio to ensure that the two contact matrices had the same total read count. Then we merged the normalized contact matrices by taking the maximum contact value at each entry in the matrices. In addition, we used a threshold to filter out noisy contacts in the adjacency matrix, i.e., all the entries with contact values <2 were removed. Next we sampled 50 spatial neighbors and 10 sequential neighbors for each sequence, resulting in a total number of 77M chromatin contacts used.

ECHO and all the baseline models were trained on a NVIDIA Tesla V100 GPU with a batch size 64 and optimized by stochastic gradient descent with a momentum of 0.9 and a learning rate of 0.5. For the loss function, we chose a mean binary cross entropy loss

$$\text{BCELoss} = -\frac{1}{N}\frac{1}{L}\sum_{i=1}^{N}\sum_{l=1}^{L}y_s^{(i)}\log(\hat{y}_l^{(i)}) + (1 - y_l^{(i)})\log(1 - \hat{y}_l^{(i)}), \tag{9}$$

where $N$ indicates the number of samples, $L$ is the length of target chromatin feature vector, $y_l^{(i)}$ and $\hat{y}_l^{(i)}$ represent the $l$-th element in target and predicted chromatin feature vector for sequence $i$, respectively.

## Attribution methods

Even though our ECHO model predicts chromatin features accurately, researchers still hope to identify DNA segments and chromatin contacts that contribute to the prediction of a specific chromatin feature. Visualizing the filters from the first convolutional layer is useful to discover DNA binding motifs, but the patterns learned by individual filters can be redundant, and one motif pattern can be the result of cooperation among multiple filters [25]. In our work, we adopt a gradient-based attribution method. The inputs to our model include both DNA sequences and their interactions, and for the simplicity, we use *gradient × input* [21] to calculate attribution scores on the inputs (DNA sequences and Micro-C contacts). The score on each base pair indicates which base pair of a DNA sequence and which chromatin contacts contribute to the prediction of a given chromatin feature. The attribution method is described in Fig 4a and explained thoroughly in the following sections.

**Attribution on chromatin contacts.** For each sequence $i$, generating its feature matrix $\Xi^{(i)}$ is a multiplication between a binary sampling matrix and the hidden representation matrix $\phi \in R^{N \times k}$, $\Xi_s^{(i)} = P_s^{(i)}\phi$ and $\Xi_c^{(i)} = P_c^{(i)}\phi$, where $P_s^{(i)} \in \{0, 1\}^{k_s \times N}$ and $P_c^{(i)} \in \{0, 1\}^{k_c \times N}$ are two binary sampling matrices used to sample sequential and spatial neighbor sequences, respectively. Both the two matrices have exactly one 1 in each row, indicating which sequence is sampled. Then we use *gradient × input* to calculate the attribution scores of the sampling matrices for chromatin feature $l$

$$[S_c^{(i)}]_l = P_c^{(i)} \odot \frac{\partial \hat{y}_l^{(i)}}{\partial P_c^{(i)}}, \quad [S_s^{(i)}]_l = P_s^{(i)} \odot \frac{\partial \hat{y}_l^{(i)}}{\partial P_s^{(i)}},$$

where $\odot$ represents the Hadamard product. $[S_c^{(i)}]_l$ and $[S_s^{(i)}]_l$ are the attribution scores of the two binary sampling matrices for chromatin feature $l$. If attribution scores need to be calculated for a set of chromatin features $\mathcal{L}$ (e.g. same type of chromatin features from different cell lines), we have

$$[S_c^{(i)}]_\mathcal{L} = \sum_{l \in \mathcal{L}}[S_c^{(i)}]_l, \quad [S_s^{(i)}]_\mathcal{L} = \sum_{l \in \mathcal{L}}[S_s^{(i)}]_l.$$

Then $[S_c^{(i)}]_l$ and $[S_s^{(i)}]_l$ are compressed to two interaction importance vectors $[V_c^{(i)}]_l \in R^{1 \times N}$ and $[V_s^{(i)}]_l \in R^{1 \times N}$ by taking the one non-zero value at each row. Each non-zero element in the vectors indicates the importance of sampling the corresponding neighbor sequence, which shows the contact importance with central sequence $i$. Since one neighbor sequence may be sampled as the sequential and spatial neighbors at the same time, we take the maximum of the two vectors to generate one single vector which is taken as the $i$-th row of a sparse interaction importance matrix $M_l \in R^{N \times N}$. An element $M_l[i, j]$ indicates the importance of a contact between sequence $j$ and the central sequence $i$ for chromatin feature $l$, while if sequence $j$ is not

sampled as $i$'s neighbor sequence, then the element is set 0. In the end, we take the absolute value and normalize each row of the interaction importance matrix as

$$\widehat{M_l}[i,j] = \frac{|M_l[i,j]|}{\max_{t \in \{1, \cdots, N\}}(|M_l[i,t]|)} \in [0, 1].$$

**Attribution on DNA sequences.**   In addition to calculating attribution scores of the central sequences like sequence-based models, ECHO computes an attribution score $[S^{(j)}]_l^{(i)}$ for the neighbor sequence $j \in \mathcal{N}^{(i)}$ for chromatin feature $l$ on the central sequence $i$ (Fig 4a)

$$[S^{(j)}]_l^{(i)} = x^{(j)} \odot \left( \frac{\partial y_l^{(i)}}{\partial \Xi^{(i)}} \cdot \frac{\partial \Xi^{(i)}}{\partial x^{(j)}} \right) \quad \forall j \in \mathcal{N}^{(i)} \cup \{i\},$$

where $\mathcal{N}^{(i)}$ indicates the neighbor set of the central sequence $i$. Similarly, to calculate the attribution scores for a set of chromatin features $\mathcal{L}$, we have

$$[S^{(j)}]_{\mathcal{L}}^{(i)} = \sum_{l \in \mathcal{L}} [S^{(j)}]_l^{(i)}.$$

The high attribution score regions in the central sequence and its neighbors significantly contribute to the prediction of the central sequence's chromatin features.

**Combining attribution on DNA sequences and chromatin contacts.**   The attribution methods on DNA sequences and Micro-C contacts are combined to further interpret ECHO, e.g., discovering important sequence patterns in the neighborhood. To identify such sequence patterns for specific chromatin features, we have two requirements. The first requirement is that the contacts which connect central sequences and the neighbor sequence need to have high attribution scores for the investigated chromatin features. Then the second one is that the sequence patterns are generated from high attribution score regions from those neighbor sequences. Therefore, we first identify highly attributed chromatin contacts according to the chromatin features investigated, and select the contacts if the investigated chromatin features are successfully predicted on the central sequence sides. Next, each pair of central and neighbor sequences connected by the contact is further attributed regarding the contributions to the chromatin features on the central sequence. To explore the TF collabortive binding mechanism, the high attribution score regions on both the central and neighbor sequences are compared with known TF motifs from JASPAR [24]. To discover important sequence patterns in the neighborhood contributing to the chromatin features on the central sequences, we collect a number of neighbor sequences satisfying the requirements discussed above and compute their attribution scores. Then sequence patterns are generated from these sequences with their attribution scores by TF-MoDISco [25].

## Supporting information

**S1 Table. Notations used in our work.**
(PDF)

**S2 Table. Comparing the mean AUROC and AUPR scores of ECHO with the baselines.**
The first three models are the baselines, the fourth model is ECHO with only spatial neighbors sampled, and the fifth model is ECHO with only sequential neighbors sampled, and the last threes are our proposed methods built on the corresponding baseline model with 10 sequential

neighbors and 50 spatial neighbors sampled.
(PDF)

**S3 Table. Comparing the performance of ECHO with ChromeGCN for 103 chromatin features on GM12878 cell line.**
(PDF)

**S1 Fig. Comparing the prediction performance between ECHO and baselines.** (a) The mean ROC curves from ECHO and three baseline models for three types of chromatin features, including TF, histone mark and DHS. ECHO achieves higher mean AUROC scores than the baselines, especially on TF and histone mark. (b) The ROC curves for each chromatin feature from ECHO and DeepCNN models. The red lines denote the median ROC curves.
(TIF)

**S2 Fig. Comparing ECHO with DeepCNN on cell-type specific chromatin feature prediction.** The results of 200 cell lines with most collected chromatin feature profiles are provided, the rest cell lines are shown in S3 Fig. The first column shows the improvement on mean AUPR score for each cell line, the second column shows the improvement on mean AUC score, the third column displays the number of collected chromatin features, and the fourth column shows the mean AUROC scores.
(TIF)

**S3 Fig. Comparing ECHO with DeepCNN on cell-type specific chromatin feature prediction.** The results of the rest 202 cell lines with least collected chromatin features are shown.
(TIF)

**S4 Fig. Scatter plots to compare the model performances of ECHO with ChromeGCN on GM12878 cell line.**
(TIF)

**S5 Fig. A box plot of average peak widths of TFs and DHSs.** The mean DHS peak width is 162bp and the mean TF peak width is 383bp.
(TIF)

**S6 Fig. Details of the graph layers and the prediction layer in ECHO.** The architectures of graph layers are varied considering the number of chromatin features, the input sequence size, and whether sequential neighbors are sampled. The model architecture reported here is for predicting 2,583 chromatin features with 50 spatial neighbors and 10 sequential neighbors per input sequence.
(TIF)

**S7 Fig. The effects of Micro-C contact distances on predicting chromatin features related with gene activation.**
(TIF)

**S8 Fig. Majority of attribution scores on Micro-C contacts are attribution scores of contacts within topologically associating domains (TADs).** The Mirco-C contacts within the first 10k sequences in Chromosome 8 are visualized in a circle. 0.988 of the total attribution scores for all chromatin features are total attribution scores of contacts within TADs, and 0.982 of the contacts are in TADs. The blue dashed lines show the hESC TAD boundaries. The black numbers on the circle index the 10k sequences, and the blue small numbers index the 22 TADs. The attribution scores of contacts for all chromatin features within each TAD are

plotted. The color transparency of the lines represents the values of attribution score.
(TIF)

**S9 Fig. Comparing original contact matrices used by ECHO and the yielded attribution matrices.** Two example regions in H1 cell line are provided. The original contact matrices are shown in the left column and the yielded attribution matrices from ECHO are shown in the right column. (a) A region where the attribution scores resemble the original chromatin contact matrices. (b) In a region where the attribution scores are different with the contact matrices, the two blue squares in the figure show the chromatin contact patterns which are not reflected by the attribution scores. The chromatin contact matrices are symmetric whereas the attribution score matrices are asymmetric with $(i, j)$−th entry denoting the importance of sampling neighbor sequence $j$ to the chromatin feature prediction on the central sequence $i$. Sum of the rows are provided on the right of each matrix. Notice that the chromatin contact matrices are populated for every 200bp sequence, but the attribution scores can be zero by default for the entire row if the 200bp sequence is not used by ECHO (ECHO only used the genomic regions with at least one TF binding events in all used cell lines, following the same strategy used by DeepSEA) or no chromatin features in H1 cell line appear in the 200bp sequence.
(TIF)

**S10 Fig. Visualization of attribution scores on DNA sequences.** (a) Attribution scores of DNA sequences for two specific TFs, JUND and CEBPB. The height of each letter (A,T,C,G) shows the attribution score for the exact base pair. The high score regions are compared with known motifs from the JASPAR database [24]. (b)Sequence patterns generated by TF-Mo-DISco [25]. For each TF, the sequence patterns are generated from the attribution scores of 100 binding sites which are also successfully predicted by ECHO. These patterns match the known binding motifs from JASPAR. (c)Attribution scores of the central sequences and the neighbor sequences toward TF binding on central sequence. The Micro-C contact values between central sequences (top) and neighbor sequences (bottom) are given. The high attribution score regions in the central sequence reflect TF binding motifs, whereas the high attribution score regions in the neighbor sequence contribute to the TF binding prediction on the central sequence. The correlated high attribution score regions reveal the potential collaborative binding mechanisms of TFs. For example, we observe a CTCF pattern in the central sequence, and a CTCF pattern and a MAX::MYC pattern in the neighbor sequence. Our observation agrees with the previous study that CTCF and MAX which frequently exist at the chromatin loop anchors may form a complex and participate in CTCF loops [39].
(TIF)

**S11 Fig. 100 TFs with the cell lines which have lowest performance improvement or perform even worse compared to DeepCNN.** TFs without known motifs from JASPAR are marked in blue, others are marked in red.
(TIF)

**S12 Fig. Differences of AUROC scores comparing ECHO with DeepCNN for TFs in multiple cell lines.** We identify ten TFs for which ECHO and DeepCNN predict quite differently among more than three cell lines. The $Y$-axes show the differences of AUROC scores (AUROC from ECHO minus AUROC from DeepCNN). TFs without known motifs are marked with '*'. (Left panels) TFs whose AUROC scores are significantly higher in ECHO than DeepCNN. (Right panels) TFs whose AUROC scores are slightly higher or lower in ECHO than DeepCNN.
(TIF)

## Author Contributions

**Conceptualization:** Zhenhao Zhang, Jie Liu.

**Data curation:** Zhenhao Zhang.

**Formal analysis:** Zhenhao Zhang, Fan Feng, Jie Liu.

**Funding acquisition:** Jie Liu.

**Investigation:** Zhenhao Zhang, Jie Liu.

**Methodology:** Zhenhao Zhang, Jie Liu.

**Project administration:** Jie Liu.

**Software:** Zhenhao Zhang.

**Supervision:** Jie Liu.

**Validation:** Zhenhao Zhang, Fan Feng, Jie Liu.

**Visualization:** Zhenhao Zhang, Fan Feng, Jie Liu.

**Writing – original draft:** Zhenhao Zhang, Fan Feng, Jie Liu.

**Writing – review & editing:** Zhenhao Zhang, Fan Feng, Jie Liu.

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
