## [Decision Letter · Decision Letter 0]

18 Jan 2022

Dear Dr. Liu,

Thank you very much for submitting your manuscript "Characterizing collaborative transcription regulation with a graph-based deep learning approach" for consideration at PLOS Computational Biology.

As with all papers reviewed by the journal, your manuscript was reviewed by members of the editorial board and by several independent reviewers. In light of the reviews (below this email), we would like to invite the resubmission of a significantly-revised version that takes into account the reviewers' comments.

All three reviewers appreciated the value of this work, but suggested a number of points, including major points, that need to be addressed in the updated version of the manuscript.

We cannot make any decision about publication until we have seen the revised manuscript and your response to the reviewers' comments. Your revised manuscript is also likely to be sent to reviewers for further evaluation.

Sincerely,

Vladimir B Teif, Ph.D.

Guest Editor

PLOS Computational Biology

Ilya Ioshikhes

Deputy Editor

PLOS Computational Biology

All three reviewers appreciated the value of this work, but suggested a number of points, including major points, that need to be addressed in the updated version of the manuscript.

Reviewer's Responses to Questions

**Comments to the Authors:**

Reviewer #1: The authors developed this new computational method called ECHO for the prediction of chromatin features by using DNA sequence and Micro-C contact maps as input. As compared to the existing method that explicitly considers the chromatin interactions, ECHO introduced computational techniques in GraphSAGE, which allows the model to be trained on graphs at a large scale, i.e. Micro-C data at high resolutions. The authors demonstrated that ECHO has improved performance for predicting chromatin features, and by applying gradient-based attribution methods, important contacts and sequence features can be identified. However, I have some concerns in terms of the performance evaluation and analysis results. See my detailed comments below:

1. One important thing that the authors should clarify in the motivation of this work is why predicting chromatin features at high-resolutions (200bp) is important. Since one of the major contributions of this work is improved scalability of the method on a large-scale chromatin interaction network, the authors should justify that this problem indeed exists. Perhaps the author could report the average width of the peaks called from these chromatin features? What's the typical size of the narrow peaks and broad peaks, can they already be captured well with the original 1kb resolution used in chromeGCN?

2. Some related works are discussed but not included in the performance comparisons. The authors briefly discussed SATORI, Basenji, and Enformer in the discussion part, but did not include any of them in the performance comparisons. The authors mainly compared ECHO against DNA sequence models with small flaking regions being considered. It is important to show that ECHO reflects the collaborative binding mechanisms among TFs that these new DNA-sequence-based models with larger receptive field sizes could not capture.

3. Some of the analysis results are not rigorous enough. For instance, the authors observed that for bins with CTCF binding events, the proportion of their associated contacts where the other anchor was also bound by CTCF increases by the genomic distance. The authors then reached the conclusion that "CTCF interactions had more effects on CTCF binding as contact distance increased". Such conclusion is strange as CTCF binding happens before the formation of CTCF loops. The main drawback of such an analysis process is that the attribute scores based on gradient only suggest "correlations" or "dependencies" not "causality". It is known that for loops or significant contacts, the ratio of CTCF loops increases with the distance. Thus, naturally, loops of longer distance would contribute more to CTCF binding predictions. The latter relationship between the distance of loops and CTCF binding is caused by the confounding factor of the relationship between the ratio of CTCF loops and distance.

Minor:

1. I found it hard to understand the following sentence "As a comparison, we computed attribution scores by using DeepCNN who could only attribute each sequence’s binding TFs to itself, so the ZBTB3 binding motif was not detected as expected." I suppose the authors mean that "DeepCNN did not utilize Micro-C contacts and thus can not identify important sequence features in its spatial neighbors?"

Reviewer #2: Summary:

In this paper, Zhang et al. aim to better characterise the collaborative activity of the epigenome mediated through 3D interactions. To achieve this, they developed the ECHO model, which leverages both DNA sequence and Micro-C data to predict the absence or presence of epigenomic features for the region of interest across different cell lines.

To evaluate ECHO's performance, the authors compared it to sequence-only models (Deep-CNN, DeepSea and DanQ) as well as one alternative model incorporating spatial information (ChromeGCN). The authors showed how ECHO out-performs all considered models, with regards to AUROC and AUPR metrics, to varying degree depending on the predicted feature of interest. More specifically, they observed a dramatic improvement for histone modifications, a moderate improvement for transcription factors and a mild improvement for Dnase hyper-sensitive sites. Nevertheless the improvement was not obvious for some underrepresented cell types. They also showed that the influence of pre-trained sequence-feature layers was only marginal, indicating the decisive contribution of the spatial components of ECHO. The authors also evaluated the influence of factors specific to chromosome-capture sequencing data and found important and varying effects on model performance depending on the linear distance-range of the interactions considered as well as on the bin size of the interaction data.

The authors also demonstrate the unique insights captured by ECHO because of its consideration for spatial interactions and functional collaborations. They manage to derive such interpretations of the model by using gradient-based attribution methods that trace the contribution of the different components of the model for individual predictions. These attribution methods allowed the authors to examine how sequence information contributed to model prediction while integrating the spatial configuration harnessed by graph layers. Of particular interest, the authors were able to reliably detect long-range CTCF-interactions as well as recover validated binding co-factors in the chromatin neighbourhood of these CTCF binding-sites. Similar properties were found for SMAD5 and H3K4me3.

Comments:

The paper presents an original and timely contribution to the field, while making relevant benchmarkings with recent instances of deep neural networks applied to epigenomic characterisation. The authors also provide a detailed description of the model and algorithm and provide the necessary supporting information online. The manuscript is also well organized and written clearly enough to be accessible to non-specialists.

Nevertheless, there are several points that I found to not be described in enough details and would justify some additional analyses before I recommend this paper for publication.

Major:

I) From the manuscript, it is unclear how redundant the sequential and spatial neighbours were. Especially given that the closest linear neighbours would tend to also be neighbours with the highest Micro-C interaction signal. Could the authors provide a quantitative evaluation of this likely redundancy? If sequential and spatial neighbours were to be extensively redundant, the benefit of the spatial components of ECHO might need to be re-evaluated. A possibly interesting extension, if we were to witness this redundancy, would be to use the observed/expected transformation of Micro-C data instead of the normalized read-counts currently used to sample spatial neighbours. Here "expected" would reflect the expected linear distance decay of Micro-C signal. The observed/expected transform might therefore more distinctly reflect the spatial configuration complementing the sequential organisation of the chromatin.

II) To further explore the merits of ECHO's spatial components, maybe more targeted and informative models should be considered when reporting model performance using the AUROC and AUPR metrics. An example that comes to mind would be to compare the original ECHO model with an ECHO model without a spatial neighbours stack, but a larger set of sequential neighbors only. For example, maybe 60 sequence neighbours split evenly up and downstream of the central sequence. This alternative model would be of a comparable complexity and highlight the exclusive contribution of the spatial stack.

III) An aspect that wasn't clear in the manuscript was the extent to which the authors were able to recover the cell-specific Micro-C pattern through the attribution method ? The reason this might be interesting is that it would indicate the extent to which the model needs to reproduce the 3D configuration of the chromatin in order to recover the main determinants of the predicted epigenomic features. What is the correspondence between spatial proximity and model relevance to predict epigenomic features? Of course such an analysis would only be relevant for the cell-lines for which Micro-C is available, but this would still be informative in further interpreting and explaining the model.

IV) It was unclear to me regarding the central regions predicted to be bound by TFs, what was the extent of the recovery of expected known motifs using the attribution methods? This is only described for CTCF, but such evaluation would constitute an important sanity check for the full set of binding factors with known motifs. If the authors could evaluate this, using for example the proportion of such central regions expected to be bound by a factor for which we also recover the corresponding motif through model attribution, it would greatly support the biological relevance of the features produced by ECHO.

Minor:

I found the comparison with the Enformer model in the Discussion section to be a bit elusive. Since Enformer, like ECHO, leverages a broader context to make its prediction, what would be the exclusive merits of ECHO if instead of predicting gene expression Enformer predicted chromatin features like ECHO?

Reviewer #3: Authors present ECHO, a graph convolutional network that leverages DNA sequence and 3D chromatin architecture derived from chromatin conformation assays to impute epigenomics assays data. This model extends on widely used sequence based models such as Basset, Basenji, and the recent Enformer by incorporating 3D genome conformation. The ideas behind this method are solid, and authors present convincing improvements on sequence based models that do not explicitly leverage 3D chromatin conformation methods, and also present better performance on a competing model that uses 3D chromatin conformation, ChromeGCN. As the deep HiC/MicroC data needed by the authors for this task is currently only limited to very few cell types, this model is more of a proof of principle but is a useful and promising extension to existing methodologies, and holds great promise for future use. Authors also make efforts to interpret the networks for biological insights and validation of known biological drivers of 3D chromatin conformation, though these efforts are sometimes anecdotal. With some polishing and better presentation of validation/interpretation analyses, this paper is a valuable contribution to the current collection of deep learning models that aim to impute genomic signal.

Major Points

ECHO’s architecture and the concept of using graph convolutions on the graph defined by significant MicroC interactions is an appropriate and refreshing use of 3D chromatin architecture for prediction of chromatin features (using the authors’ term) is satisfying. However, I do not understand why sequential neighbors are necessarily included? Presumably short and long range interacting regions that aid the prediction task will be included as per MicroC signal (spatial neighbors). Can the authors explicitly describe the justification of inclusion of ‘sequential neighbors’ on top of ‘spatial neighbors’? One would expect spatial neighbor set to include sequential neighbors. Does the performance of the model suffer dramatically if only ‘spatial neighbors’ are used?

Contact distance stratified models presented in Figure 2 are interesting. For histone mark prediction, is it possible that histone marks that tend to exhibit broader peaks tend to benefit from short range contacts?

The upsampling strategy in “Chromatin contact resolution…” section seems somewhat problematic. More specifically, assuming all 200bp bins will have the same upsampled interaction frequency derived from 1kb Hi-C map is wrong, as I am sure the authors are also aware of. It is clear this strategy is used in this analysis to demonstrate the reduced performance when 200bp resolution interactions are derived from 1kb (HiC) resolution interactions. However, since ECHO uses a sampling strategy for spatial neighbors, would this not mean five 200bp bins that cover a 1kb (HiC resolution) bin with high interaction frequency will be overly represented in the graph structure? In this case, I am unsure if poorer performance results from reduced resolution vs. biased sampling; or these two related effects may be intertwined. I would recommend an alternative analysis; it seems more sensible to me to collapse 200kb MicroC to 1kb resolution without needing any sampling and show that model suffers at this resolution, unless the model is strictly tuned to 200bp resolution. Alternatively, maybe the sampling strategy for lower resolution may be altered.

The results showing the enrichment of long range contacts that are attributed to CTCF motif is convincing and a reasonable validation of ECHO’s ability to leverage 3D structure in imputing signal. Can a simpler CTCF loops based enrichment be also performed, using loop calls from a loop caller such as HiCCUPS or ChIA-Pet data?

In line 238, authors state “In this experiment, we investigated whether specific sequence patterns in the neighborhood existed..”. Are they referring to spatial or sequential neighborhood, or both?

Minor points

Authors claim “Motivated by this, most current computational models characterize TF binding and other chromatin features only from the DNA sequences”. Hardly true, as ATAC/DNA-seq, and variations of ChIP-seq are also leveraged in models. I believe authors are referring to a specific subset of computational models that leverage DL - such as Basset, Basenji, Enformer and others – and aim to impute/predict signal. This needs to be clarified.

Paragraph 2 in intro (line 33) hardly follows the flow and is a hard transition for audiences that are not experts in this domain.

Line 39, 42: contact map concept has not been introduced or explained at all.

Line 46: What is a chromatin feature? It sounds like it is various properties of chromatin (Histone Mods, TF binding and chromatin accessibility) as measured by epigenomics/TF binding ChIP-seq assays and chromatin accessibility assays. Please be more specific.

Line 218, are the candidate cis-regulatory elements referred here ccREs from the ENCODE consortium? Please clarify/cite.

Figure 5a, the arc between interacting regions is partially missing. Please correct.

**Have the authors made all data and (if applicable) computational code underlying the findings in their manuscript fully available?**

Reviewer #1: Yes

Reviewer #2: Yes

Reviewer #3: Yes

PLOS authors have the option to publish the peer review history of their article (what does this mean?). If published, this will include your full peer review and any attached files.

Reviewer #1: No

Reviewer #2: **Yes: **Vipin Kumar

Reviewer #3: No
---

## [Decision Letter · Decision Letter 1]

2 May 2022

Dear Dr. Liu,

We are pleased to inform you that your manuscript 'Characterizing collaborative transcription regulation with a graph-based deep learning approach' has been provisionally accepted for publication in PLOS Computational Biology.

Best regards,

Vladimir B Teif, Ph.D.

Guest Editor

PLOS Computational Biology

Ilya Ioshikhes

Deputy Editor

PLOS Computational Biology

Reviewer's Responses to Questions

**Comments to the Authors:**

Reviewer #1: The authors have addressed all my concerns in the current version.

Reviewer #2: The authors have addressed the points regarding the examination of additional models to further benchmark ECHO. In particular the addition of the wider sequential model and the exclusively spatial model strengthen the necessity to account for both components (sequential and spatial) of the original ECHO model.

The examination of the extent to which attribution scores could relate to Micro-C data is a welcomed addition and highlights the dynamic manner with which the model leverages the 3D context for its prediction.

The examination of the recovery of motifs for a collection of TFs is satisfactory. I would still be interested in evaluating the recovery of motifs at regions where we expect to find TF-binding events based on publicly available ChIP-seq data for some of the cell-lines presented in the study. But since the authors provide the code necessary to reproduce the analysis, this poses no critical issue.

The revised discussion section comparing ECHO with Enformer, Basenji and ChromeGCN is much more compelling and highlights much more clearly the distinct properties of ECHO (range-free interacting neighbours and grid-based 1D convolution of the 3D neighbourhood).

In light of these revisions I can recommend this manuscript for publication.

Reviewer #3: Thank you for your responses to my criticisms.

**Have the authors made all data and (if applicable) computational code underlying the findings in their manuscript fully available?**

Reviewer #1: None

Reviewer #2: Yes

Reviewer #3: Yes

PLOS authors have the option to publish the peer review history of their article (what does this mean?). If published, this will include your full peer review and any attached files.

Reviewer #1: No

Reviewer #2: **Yes: **Vipin Kumar

Reviewer #3: No

---

## [Editor Report · Acceptance letter]

1 Jun 2022

PCOMPBIOL-D-21-02056R1 

Characterizing collaborative transcription regulation with a graph-based deep learning approach

Dear Dr Liu,

I am pleased to inform you that your manuscript has been formally accepted for publication in PLOS Computational Biology. Your manuscript is now with our production department and you will be notified of the publication date in due course.

With kind regards,

Zsofia Freund
